# Prospective, Observational Study of Aflibercept Use in Combination with FOLFIRI in Patients with Metastatic Colorectal Cancer: A Real-World Effectiveness Study

**DOI:** 10.3390/cancers16111992

**Published:** 2024-05-24

**Authors:** Agnieszka Durbajło, Marcin Świeżyński, Beata Ziemba, Danuta Starzyczny-Słota, Marzena Samborska-Plewicka, Anna Cencelewicz-Lesikow, Agata Chrzanowska-Kapica, Aneta Dobrzyńska-Rutkowska, Iwona Drab-Mazur, Monika Kulma-Kreft, Magdalena Sikora-Skrabaka, Elwira Matuszewska, Małgorzata Foszczyńska-Kłoda, Tomasz Lewandowski, Grzegorz Słomian, Krystyna Ostrowska-Cichocka, Ewa Chmielowska, Rafał Wiśniowski, Anna Twardosz, Katarzyna Wierzbicka, Leszek Rumianowski, Lucjan Wyrwicz

**Affiliations:** 1Oncology and Chemotherapy Clinic, Maria Sklodowska-Curie Memorial Research Institute of Oncology, Roentgena 5, 02-781 Warsaw, Poland; agnieszka.durbajlo@nio.gov.pl (A.D.);; 2Prof. Franciszek Łukaszczyk Memorial Centre of Oncology, 85-796 Bydgoszcz, Poland; 3Lower Silesia Centre of Oncology, 53-413 Wrocław, Poland; ba.ziemba@wp.pl; 4Maria Sklodowska-Curie Memorial Research Institute of Oncology, 44-102 Gliwice, Poland; 5Memorial of Jan of Dukla Oncology Centre of Lublin County, 20-090 Lublin, Polandatwardosz@cozl.eu (A.T.); 6Memorial of Zofia from Zamojski Family Tarnowska Voivodeship Hospital, 39-400 Tarnobrzeg, Poland; imazur@sand.pl; 7PCK Memorial Maritime Hospital, 81-519 Gdynia, Poland; 8Voivodeship Specialist Hospital, 41-902 Bytom, Poland; magda_sikora19@wp.pl; 9Maria Sklodowska-Curie Memorial Oncology Centre, 15-027 Białystok, Poland; ematuszewska@onkologia.bialystok.pl (E.M.);; 10Western Pomerania Oncology Centre, 71-730 Szczecin, Poland; mfoszczynska@onkologia.szczecin.pl; 11Radom Heroes of June 76 Memorial Radom Oncology Centre, 26-600 Radom, Poland; tomasz.lewandowski@poliklinika.net; 12Specialist Voivodeship Hospital nr 3, 44-200 Rybnik, Poland; gslomian.szpital.rybnik@gmail.com; 13Specialist Oncology Hospital Nu-Med, 97-200 Tomaszów Mazowiecki, Poland; 14The Beskids Oncology Centre, 43-300 Bielsko-Biała, Poland; 15Department of Oncology and Radiotherapy, University Clinical Centre, 80-952 Gdańsk, Poland; 16Department and Clinic of Oncology, Medical University, 60-569 Poznań, Poland; leszek.rumianowski@skpp.edu.pl

**Keywords:** metastatic colorectal cancer, aflibercept, FOLFIRI, second-line treatment, mCRC, antiangiogenics

## Abstract

**Simple Summary:**

The VELOUR randomized phase III trial established aflibercept (an anti-angiogenic targeting VEGF-A, VEGF-B, and placenta growth factor) combined with FOLFIRI as an effective treatment in patients with metastatic colorectal cancer (mCRC) failing a prior oxaliplatin-based regimen. This real-world study mandated by Polish Health Authorities aimed at assessing the benefits/risks of aflibercept plus FOLFIRI prescribed to Polish patients with mCRC according to reimbursement criteria. The activity of aflibercept plus FOLFIRI in such patients was confirmed, and no new safety signal was reported.

**Abstract:**

Background: This was an observational study prospectively evaluating the effectiveness and safety of aflibercept/FOLFIRI administered in second-line mCRC per the reimbursement criteria in Poland. Methods: Consecutive mCRC patients who progressed with first-line oxaliplatin-based chemotherapy received aflibercept (4 mg/kg IV) followed by FOLFIRI every 2 weeks until progression or unacceptable toxicity. The primary endpoint was progression-free survival (PFS); overall survival (OS) and safety were the secondary endpoints. Results: A total of 93 patients were treated at 17 Polish sites. A median of 10 cycles was administered. Over a median treatment duration of 5.3 months, median PFS and median OS were 8.4 months [95% CI, 6.9–9.9] and 27.0 months [95% CI, 23.9–30.1], respectively. There was no significant impact of primary tumor location, metastatic site, or KRAS status on PFS and OS. Main grade ≥ 3 adverse events were neutropenia (16%), hypertension (8%), diarrhea (4%), and stomatitis (4%). Conclusions: The benefits/risks of Aflibercept plus FOLFIRI administered per the Polish reimbursement criteria in second-line treatment of mCRC after failure of a prior oxaliplatin-based regimen is confirmed.

## 1. Introduction

Colorectal cancer (CRC) is among the most frequent tumor types in Western countries and represents the second cause of cancer-related mortality worldwide, with 935,000 deaths in 2020 [1,2]. In Poland, CRC represents the third most common cause of cancer mortality, with age-standardized mortality rates of 23 and 17 per 100,000 in men and women, respectively [3]. If detected at an early stage, approximately 90% of patients will survive at 5 years [4,5]. However, the prognosis of advanced disease is poor, with a 5-year overall survival (OS) rate of only 13% [6].

First-line treatment of metastatic CRC (mCRC) typically combines oxaliplatin or irinotecan with fluoropyrimidines (5-fluorouracil or capecitabine) and targeted therapies (anti-EGFR agents [cetuximab or panitumumab] for RAS/BRAF wild-type tumors of the left colon or bevacizumab in other cases) [7,8,9,10,11,12,13,14,15]. At first disease progression, patients usually receive the alternative chemotherapy combined with an anti-VEGF agent [16,17,18]. Targeted therapies in combination with chemotherapy may lead to improved tumor response rates, facilitating resection of distant metastases, especially in the liver [19]. 

Aflibercept is a recombinant fusion protein which comprises key domains of human vascular endothelial growth factor receptor (hVEGFR)-1 and hVEGFR-2 fused to the Fc portion of human immunoglobulin G1 [20]. Compared to bevacizumab, which only binds to VEGF-A, aflibercept more potently inhibits VEGF-A and also neutralizes VEGF-B and placental growth factor (PIGF1 and PIGF2), with a long circulating half-life [21]. In preclinical models, aflibercept inhibited tumor growth, either alone or combined with chemotherapy, including 5-fluorouracil or irinotecan [22].

In phase I and early phase II trials, aflibercept was active in a range of tumors, including CRC, and showed increased responses when combined with chemotherapy [23]. The aflibercept dose of 4 mg/kg given every 2 weeks was selected for development because it provided an adequate aflibercept free-to-bound ratio at the end of a 2-week dosing interval at steady state. 

Registration in Europe was based on the VELOUR double-blind, placebo-controlled, phase III trial, which enrolled 1226 patients with mCRC who failed a prior oxaliplatin-based regimen [23,24]. Compared to the placebo/FOLFIRI treatment, the aflibercept/FOLFIRI treatment significantly improved overall survival (OS; 13.5 vs. 12.1; HR = 0.817 [95% CI, 0.71–0.94], *p* = 0.0032) and the primary end-point of the trial, and this was associated with a significantly higher PFS (6.9 vs. 4.7 months; HR = 0.76 [95% CI, 0.58–0.99], *p* = 0.00007) and tumor objective response rate (ORR; 19.8% vs. 11.1%, *p* = 0.0001). The survival benefit was confirmed in the long-term with a survival probability at 2 years of 28.0% vs. 18.7% in the aflibercept and placebo arms, respectively [25]. Notably, VELOUR was the unique phase III trial conducted in a second-line setting that enrolled patients who rapidly failed (within 6 months) prior adjuvant oxaliplatin, a subset of patients known to have a particularly poor prognosis [26]. There was no interaction between prior bevacizumab exposure (*n* = 373) and the treatment, but there was a significant interaction indicating a greater treatment effect for patients with liver-only metastases (*n* = 299) compared to those having extra-hepatic disease (associated or not with liver metastases) [25]. The most common grade 3–4 adverse events (AEs) in the aflibercept plus FOLFIRI arm included neutropenia (36.7% vs. 29.5%), diarrhea (19.3% vs. 7.8%), hypertension (19.1% vs. 1.5%), asthenic conditions (16.9% vs. 10.6%), stomatitis/ulceration (13.7% vs. 5.0%), infections (12.3% vs. 6.9%), and proteinuria (7.9% vs. 1.2%) [23]. Prior treatment with bevacizumab did not appear to significantly impact the safety profile of aflibercept [27].

The prospective observational study presented here was mandated by health authorities and was aimed at confirming the benefits/risks of aflibercept plus FOLFIRI given in second-line treatment after failure of an oxaliplatin-based regimen in Polish patients with mCRC in daily clinical practice.

## 2. Materials and Methods

### 2.1. Study Design

This study planned to enroll 16 to 18 sites in Poland. All the participating clinics and medical practices had experience in the management of mCRC patients and were willing to prescribe aflibercept plus FOLFIRI according to the Polish reimbursement criteria. Each participating physician was asked to include consecutive patients to limit bias related to physician-led patient selection. A screening form was implemented at each site to document patient enrolment and to capture why an eligible patient was not included.

There was no random selection of investigators in this registry in order to not compromise the recruitment timelines. To avoid bias due to potential specific site(s) effects, a maximum of 15 patients were allowed per site. Moreover, special attention was paid to ensure that the participating sites in the registry covered the whole country. The patients received aflibercept (4 mg/m^2^ of body weight administered as intravenous (iv) infusion) followed by FOLFIRI (irinotecan 180 mg/m^2^ iv plus leucovorin 400 mg/m^2^ on day 1 and fluorouracil [iv bolus of 400 mg/m^2^ followed by a continuous infusion of 2400 mg/m^2^ over 46 h). This represented one cycle, and each treatment cycle was repeated every 2 weeks. Treatment was administered until disease progression or unacceptable toxicity. This study was approved by the Institutional Review Board at each center and was conducted in compliance with the principles of the Declaration of Helsinki and Good Clinical Practice guidelines. 

### 2.2. Patient Selection

Eligible patients were aged at least 18 years old, had mCRC progressing during or after an oxaliplatin-based regimen (only one prior metastatic regimen was allowed), had an Eastern Cooperative Oncology Group performance status (ECOG PS) 0–1, had signed an informed consent form, were willing to comply with protocol requirements, and were receiving appropriate contraception measures. Patients with disease progression within 6 months after adjuvant treatment were not allowed. The exclusion criteria included prior therapy with irinotecan, hypersensitivity to the active substance or any excipient, known Gilbert’s syndrome, dihydropyrimidine dehydrogenase deficiency, and any planned other type of systemic therapy, radiotherapy, or surgery.

### 2.3. Assessment

The disease history (including tumor location and stage, KRAS/NRAS/BRAF status, prior treatments received), patient characteristics, exposure to aflibercept and FOLFIRI (total dose, number of cycles, duration of treatment, dose reductions and dose delay, reasons for treatment discontinuation), duration of follow-up, and adverse events were collected. The tumor objective response rate (ORR) was defined by the percentage of patients with a complete response (CR) or partial response (PR) per investigator judgment. The disease control rate (DCR) was defined as the percentage of patients with CR or PR or stable disease (SD) with the best response to the therapy. PFS was defined as the time from the first cycle initiation to the date of the first documented disease progression or death. OS was defined as the time from the first cycle initiation to the date of death from any cause.

### 2.4. Statistics

Statistical analyses were descriptive, and *p*-values were exploratory. Therefore, no formal sample size calculation was performed. It was estimated that around 100 patients with mCRC would provide sufficient precision in the results. OS and PFS were estimated using the Kaplan–Meier method. Effectiveness was analyzed in all the patients enrolled, and safety was analyzed in all the patients exposed to at least one dose of aflibercept. Continuous data were presented as means (SD). Categorical data were presented as absolute numbers with percentages. 

## 3. Results

### 3.1. Baseline Characteristics and Treatment Exposure

Overall, 93 patients with mCRC were enrolled at 17 Polish sites between August 2018 and August 2021. Their baseline clinical characteristics are detailed in Table 1. Of the 93 patients enrolled in the study, 55 (59.1%) were males, and the median age was 62 years (range from 29 to 81). The tumors were mainly located in the left colon (81.7%), and 69.8% of the patients had synchronous metastases. KRAS, NRAS, and BRAF mutations were detected in 41.9%, 3.2%, and 4.3% of the patients, respectively. The primary tumor was resected in most of the patients (88.2%). A total of 39 patients received prior adjuvant chemotherapy, and 24 patients had a prior resection of metastases. All the patients received prior FOLFOX chemotherapy for a median of 11 cycles, and 23 patients received a prior EGFR inhibitor. 

Information regarding the treatment exposure with aflibercept plus FOLFIRI is provided in Table 2. Overall, the patients received 1066 cycles of aflibercept plus FOLFIRI, corresponding to a median of 10 cycles (range 1–38) and a median duration of exposure of 5.3 months. Abnormal laboratory tests and/or adverse events resulted in a dose delay for 170/1066 cycles (16%). An aflibercept dose reduction was reported for 18 cycles (1.7%), and dose omission was reported for 16 cycles (1.5%). Before each cycle, a premedication was administered, including atropine (83% of patients), ondansetron (96%), metoclopramide (7%), dexamethasone (70%), hydrocortisone (3%), and thiethylperazine (10%). At the end of the study, the treatment was still ongoing in 1 patient, and 92 patients had discontinued therapy, mainly due to disease progression (51.5%), patient request (15.2%), or adverse events (12.1%). 

### 3.2. Efficacy

The median duration of follow-up was 12 months. There were 76 patients with progression or death events occurring during the treatment period. The median PFS (primary endpoint), was 8.4 months [95% CI, 6.9–9.9] (Figure 1). At one year, 32.9% [95% CI, 22.9–44.2] of the patients had not progressed. 

Over a median 12-month follow-up period, there were 18 death events, with a median survival of 27.0 months (95% CI, 23.9–30.1) and a survival rate of 78.7% [95% CI, 66.3–88.1] at 12 months (Figure 2). 

PFS and OS did not significantly differ according to the primary tumor location [Figure 3 and Figure 4], the location of metastases [Figure 5 and Figure 6], or the presence or absence of KRAS mutations [Figure 7 and Figure 8]. 

### 3.3. Safety

An overview of adverse events is provided in Table 3 and Table 4. Of the 93 patients exposed to aflibercept plus FOLFIRI, 71 patients (76%) experienced 627 adverse events (AEs) of any grade, and 32 patients had grade ≥ 3 AEs (34%) according to the WHO Toxicity Grading Scale (Table 4). The most common adverse events were neutropenia (29 patients, 31%), diarrhea (23 patients, 25%), hypertension (19 patients, 20%), asthenia (17 patients, 18%), proteinuria (15 patients, 16%), and stomatitis (14 patients, 15%). The AEs were of grade 1–2 according to the WHO Toxicity Grading Scale in most cases (67%). 

Serious AEs, regardless of causality, were reported by 13 patients (14%) (see Appendix A). Two patients had pulmonary embolisms (assessed to be not related to aflibercept; both patients recovered), and one had reversible toxic encephalopathy syndrome. Overall, 26 patients experienced adverse events leading to treatment discontinuation. One patient aged 77 years with history of controlled hypertension experienced a stroke leading to death 3 weeks after the 4th cycle of aflibercept. The tolerability of aflibercept plus FOLFIRI was in agreement with the known safety profile of the regimen [24,25].

## 4. Discussion

This prospective observational study requested by Polish Health Authorities evaluated the outcomes of treatment with aflibercept combined with FOLFIRI in patients with mCRC who failed an oxaliplatin-based regimen according to the Polish reimbursement criteria. Overall, 17 major cancer centers reflecting daily clinical practice in Poland recruited 93 patients. All the patients were managed in accordance with the summary of product characteristics and the current drug program for CRC in Poland. In general, the results of pivotal phase III trials are the most important in the process of new product registration. However, registration trials have limitations, since the patients enrolled have to satisfy stringent eligibility criteria which may not reflect daily clinical practice [28]. For this reason, the results of post-approval clinical trials conducted in everyday clinical practice are important for reimbursement purposes. 

Almost half of the patients enrolled in this prospective observational study had an ECOG PS 2; thus, the results appear to be complementary to the VELOUR phase III trial which almost exclusively included patients with an ECOG 0–1 [24]. In this less-selected Polish population with 42% KRAS mutant patients, the activity of aflibercept plus FOLFIRI appears to be even better than in the VELOUR trial in terms of PFS (6.9 months in VELOUR versus 8.4 months in our cohort) and OS (13.5 months in VELOUR versus 27.0 months in our cohort). Moreover, similarly to VELOUR, the efficacy did not appear to be influenced by primary tumor location, metastatic site, or RAS status [24,25,29]. The safety profile of aflibercept plus FOLFIRI in our study also appeared to be in agreement with the safety profile of the VELOUR trial. Main grade ≥ 3 adverse events included hypertension (8%), diarrhea (4%), and stomatitis (4%), and no unexpected safety signal was reported.

Our results are also in agreement with those of a large prospective, observational study (QoLiTrap) which included 1277 patients treated with aflibercept plus FOLFIRI in daily clinical practice in Germany, Austria, and Switzerland [30]. Half of them were RAS mutant, and 6.7% had an ECOG PS of 2 or 3. In a second-line setting, this rather fit population showed a median OS of 15.2 months, a median PFS of 8.1 months, and an ORR of 24.0%. Similarly to our cohort, there was no influence of tumor sidedness or RAS status on the treatment outcomes. Interestingly, QoliTrap also documented the activity of aflibercept/FOLFIRI in 168 patients previously treated with EGFR inhibitors, and the outcomes appeared to be even better in this subgroup of patients, with a median OS of 17.4 months, a median PFS of 9.4 months, and an ORR of 23.7%. Such patients were not enrolled in VELOUR, and only a few patients received prior EGFR inhibitors in our cohort. Lastly, most of the patients enrolled in QoliTrap fulfilled the EORTC-QLQ-C30 questionnaire, and it is reassuring to see that quality of life was maintained during the aflibercept/FOLFIRI treatment.

This study has limitations. First, it enrolled patients who were less fit than in the VELOUR phase III trial. Second, the timing of follow-up visits and tumor assessments reflected the daily practice of Polish physicians, and there was no central review of imaging. These factors may have affected the evaluation of PFS, but no major differences were observed in terms of PFS and OS compared to VELOUR [24]. Third, the protocol did not plan to collect ORR and health-related quality of life data. Lastly, the safety profile should be interpreted with caution due to possible underreporting in a real-world setting.

## 5. Conclusions

In conclusion, we found that the aflibercept in combination with FOLFIRI regimen is a beneficial therapeutic option for mCRC patients when administered as a second-line treatment after failure of an oxaliplatin-based regimen. It shows a manageable toxicity and good efficacy, defined as underlying disease stabilization and substantial progression-free and overall survival.

## Figures and Tables

**Figure 1 cancers-16-01992-f001:**
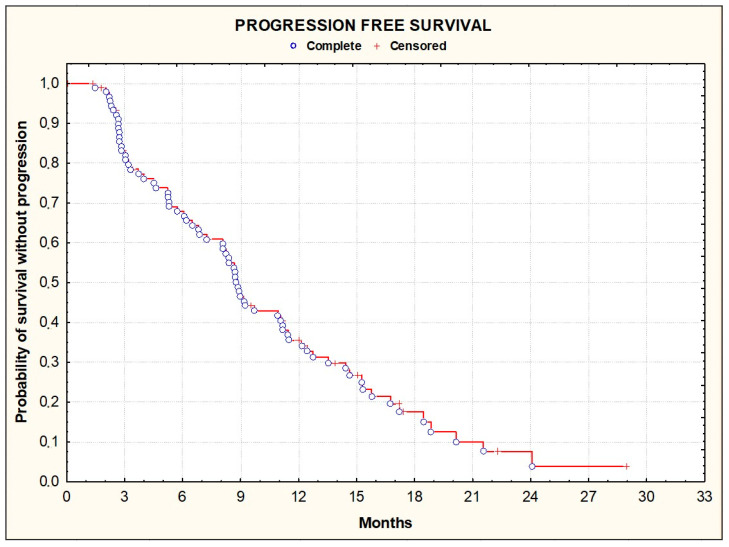
Kaplan–Meier plots illustrating progression-free survival among the enrolled patients.

**Figure 2 cancers-16-01992-f002:**
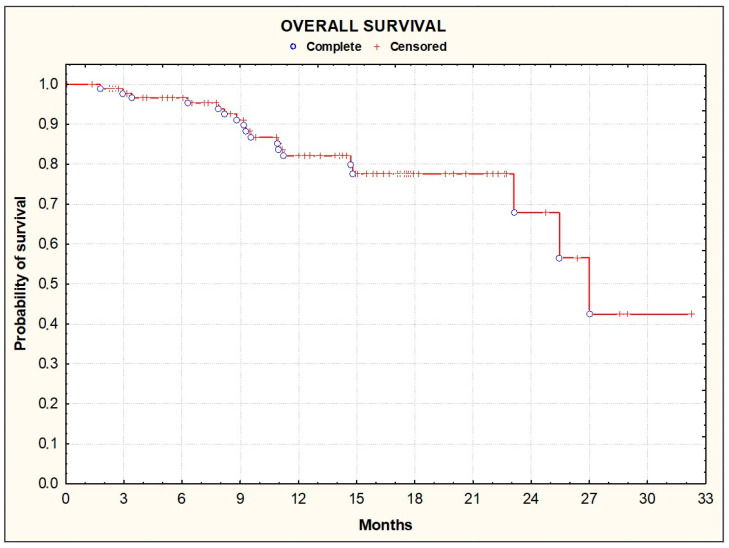
Kaplan–Meier plots illustrating overall survival among the enrolled patients.

**Figure 3 cancers-16-01992-f003:**
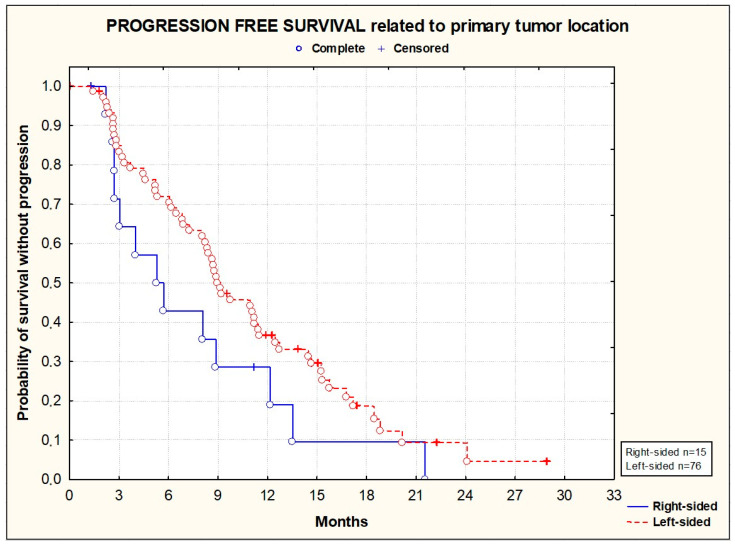
Kaplan–Meier plots illustrating progression-free survival according to primary tumor location (*p* = 0.1072; Wilcoxon test).

**Figure 4 cancers-16-01992-f004:**
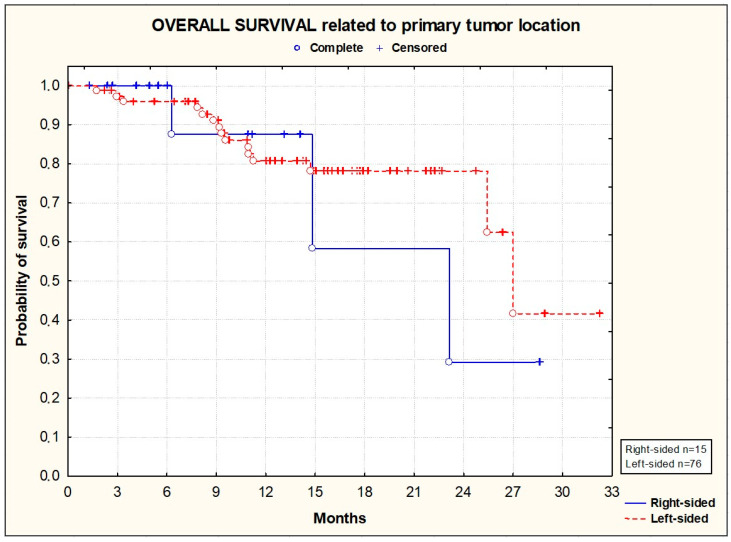
Kaplan–Meier plots illustrating overall survival according to primary tumor location (*p* = 0.9694; Wilcoxon test).

**Figure 5 cancers-16-01992-f005:**
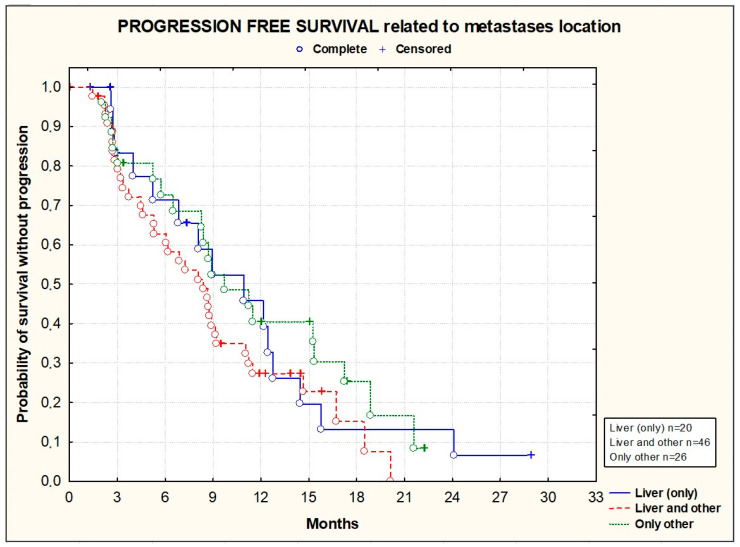
Kaplan–Meier plots illustrating progression-free survival according to primary metastasis location (*p* = 0.4330; Chi^2^ test).

**Figure 6 cancers-16-01992-f006:**
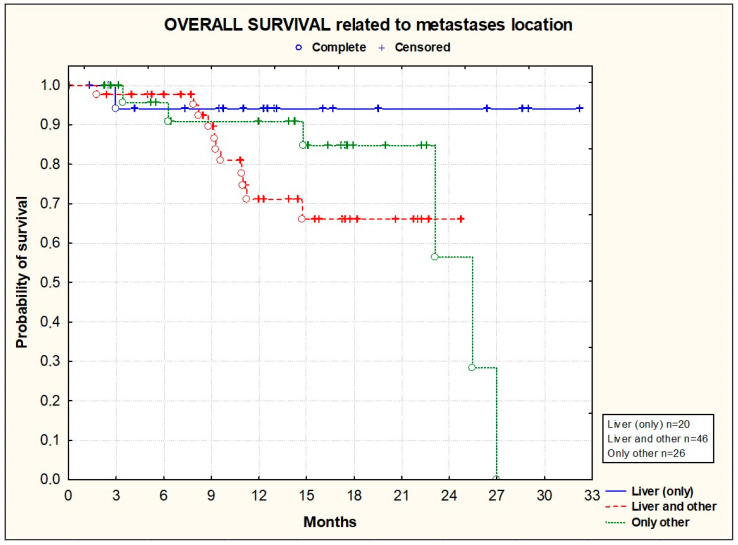
Kaplan–Meier plots illustrating overall survival according to metastasis location (*p* = 0.2265; Chi^2^ test).

**Figure 7 cancers-16-01992-f007:**
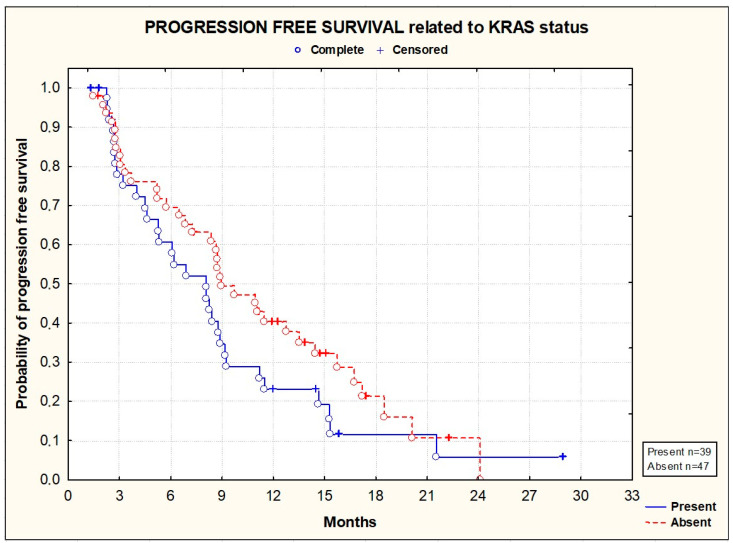
Kaplan–Meier plots illustrating progression-free survival according to KRAS status (*p* = 0.1637; Gehan–Wilcoxon test).

**Figure 8 cancers-16-01992-f008:**
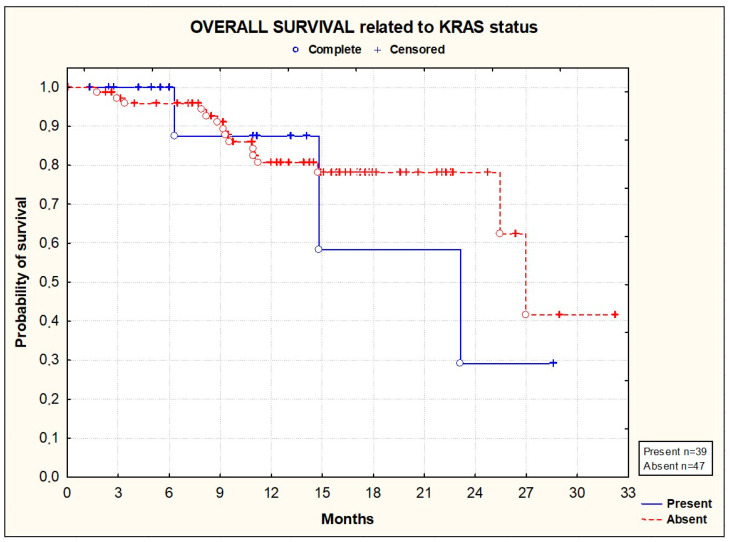
Kaplan–Meier plots illustrating overall survival according to KRAS status (*p* = 0.4045; Gehan–Wilcoxon test).

**Table 1 cancers-16-01992-t001:** Baseline clinical characteristics of the patients.

Characteristic	Overall Population (*n* = 93)
Males, *n* (%)	55 (59.1%)
Median age (range)	62 (29–81)
Primary tumor, *n* (%)	
Right-sided	15 (16.1)
Resected	24 (32.0)
Metastases, *n* (%)	
Synchronous	60 (64.5)
Prior resection of metastases	24 (32.0)
Metastatic sites at inclusion, *n* (%)	
Liver	66 (71.0)
Lung	46 (49.5)
Lymph node	38 (40.9)
Peritoneum	11 (11.8)
Other	26 (28.3)
Liver-only disease	20 (21.7)
RAS/BRAF status, *n* (%)	
KRAS	39 (42%)
NRAS	3 (3%)
BRAF	4 (4%)
Prior therapies, *n* (%)	
Adjuvant chemotherapy	39 (49.4)
Radiotherapy	22 (23.7)
Oxaliplatin-based chemotherapy	93 (100)
EGFR inhibitors	23 (24.7%)
ECOG performance status at inclusion	
1	51 (54.8%)
2	42 (45.2%)

**Table 2 cancers-16-01992-t002:** Modalities of treatment with FOLFIRI and aflibercept.

Modalities of Treatment with Aflibercept + FOLFIRI	
Duration of treatment, months	*n* = 93
Mean ± SD	6.3 ± 5.0
Range	0–22.8
Median	5.0
Q1–Q3	2.4–8.5
Number of cycles	*n* = 93
Mean ± SD	11.5 ± 8.2
Range	1.0–38.0
Median	10
Q1–Q3	5.0–15.0
Aflibercept treatment	
Dose administered per cycle, mg/kg	*n* = 93
Mean ± SD	3.99 ± 0.1
Range	2.99–4.32
Median	4.0
Q1–Q3	4.0–4.0
Total dose administered per cycle, mg	*n* = 93
Mean ± SD	314.9 ± 74.2
Range	194–568
Median	300
Q1–Q3	256–368
Total (cumulative) dose administered, mg	*n* = 93
Mean ± SD	3427 ± 2774
Range	284–4336
Median	3124
Q1–Q3	1400–4336
Fluorouracil treatment	
Total dose administered per cycle, mg	*n* = 93
Mean ± SD	4451 ± 1433
Range	612–7248
Median	4816
Q1–Q3	3870–5516
Irinotecan treatment	
Total dose administered per cycle, mg	*n* = 93
Mean ± SD	331.7 ± 44.5
Range	200.0–466.0
Median	338.0
Q1–Q3	306.0–360.0
Folinic acid treatment	
Total dose administered per cycle, mg	*n* = 93
Mean ± SD	503.3 ± 227.9
Range	154.7–1035.0
Median	400.0
Q1–Q3	340.0–730.0

**Table 3 cancers-16-01992-t003:** Overview of adverse events.

Patients, N (%)	Safety Population (*N* = 93)
N of patients reporting AEs of any grade	71 (76%)
N of patients reporting grade ≥ 3 AEs	32 (34%)
N of patients reporting serious AEs	13 (14%)
N of patients reporting AEs leading to Tx discontinuation	26 (28%)
N of patients reporting AEs leading to death	1 (1%)

AEs: adverse events; Tx: treatment.

**Table 4 cancers-16-01992-t004:** List of reported adverse events and their grades according to the WHO Toxicity Grading Scale * (safety population).

N of Patients with Adverse Events	All Grade *N* (%)	G 1–2 *N* (%)	G 3–4 *N* (%)	G 5 *N* (%)
N of patients reporting any adverse event	71 (76%)	62 (67%)	32 (34%)	1 (1%)
Most common adverse events				
Diarrhea	23 (25%)	19 (20%)	4 (4%)	-
Hypertension **	19 (20%)	11 (12%)	7 (8%)	-
Asthenia	17 (18%)	15 (16%)	2 (2%)	-
Stomatitis	14 (15%)	10 (11%)	4 (4%)	-
Abdominal pain, upper	12 (13%)	11 (12%)	1 (1%)	-
Decreased appetite **	8 (9%)	7 (8%)	-	-
Upper airway infection	8 (9%)	8 (9%)	-	-
Vomiting	6 (6%)	5 (5%)	1 (1%)	-
Constipation	6 (6%)	6 (6%)	-	-
Weight decrease	6 (6%)	6 (6%)	-	-
Dysphonia	6 (6%)	6 (6%)	-	-
Epilation	6 (6%)	5 (5%)	1 (1%)	-
Nausea	5 (5%)	5 (5%)	-	-
Headache	5 (5%)	5 (5%)	-	-
Chest pain	5 (5%)	5 (5%)	-	-
Tachycardia	4 (4%)	4 (4%)	-	-
Epistaxis	4 (4%)	4 (4%)	-	-
Esophageal candidiasis	4 (4%)	2 (2%)	2 (2%)	-
Erythrodysesthesia syndrome	4 (4%)	4 (4%)	-	-
Mouth ulceration	3 (3%)	3 (3%)	-	-
Pyrexia	3 (3%)	3 (3%)	-	-
Polyneuropathy	3 (3%)	3 (3%)	-	-
Joint swelling	3 (3%)	3 (3%)	-	-
Acute sinusitis	3 (3%)	3 (3%)	-	-
Skin lesions	3 (3%)	3 (3%)	-	-
Laboratory tests				
Neutropenia	29 (31%)	14 (15%)	15 (16%)	-
Proteinuria	15 (16%)	12 (13%)	3 (3%)	-
Thrombocytopenia	12 (13%)	12 (13%)	-	-
Leukopenia	9 (10%)	8 (9%)	1 (1%)	-
Hematuria	8 (9%)	8 (9%)	-	-
Alkaline phosphatase increase	6 (6%)	6 (6%)	-	-
Anemia of malignant disease	5 (5%)	4 (4%)	1 (1%)	-
ALAT increase	5 (5%)	5 (5%)	-	-
ASAT increase	5 (5%)	5 (5%)	-	-
Glutamyl transferase increased	5 (5%)	5 (5%)	-	-
Hyperbilirubinemia	5 (5%)	5 (5%)	-	-
Hypokalemia	4 (4%)	3 (3%)	1 (1%)	-
Hyponatremia	3 (3%)	3 (3%)	-	-

* 1–5 grades according to WHO Toxicity Grading Scale; ** single event with no grading.

## Data Availability

Qualified researchers may request access to patient-level data and related study documents including the clinical study report, study protocol with any amendments, blank case report form, statistical analysis plan, and dataset specifications. Patient-level data will be anonymized, and study documents will be redacted to protect the privacy of our trial participants. Further details on Sanofi’s data sharing criteria, eligible studies, and process for requesting access can be found at: https://www.vivli.org/ (accessed on 23 May 2024). Sanofi does not proactively post observational studies on Vivli. Instead, there is a mechanism for researchers to request access to the study data on Vivli through an enquiry form.

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
