# Peer review of "Prospective, Observational Study of Aflibercept Use in Combination with FOLFIRI in Patients with Metastatic Colorectal Cancer: A Real-World Effectiveness Study"

_cancers, 2024, doi:10.3390/cancers16111992_

Round 1

Reviewer 1 Report

Comments and Suggestions for Authors

In this manuscript, the authors evaluated the effectiveness and safety of aflibercept plus FOLFIRI in the treatment of mCRC patients in daily clinical practice. Clinical results showed that aflibercept in combination with FOLFIRI regimen is a beneficial therapeutic option for mCRC patients when administered as second-line treatment after failure of an oxaliplatin-based regimen, with a manageable toxicity and good efficacy defined as underlying disease stabilization and substantial progression free and overall survival. Furthermore, no significant differences in clinical outcomes were confirmed regardless of primary tumor location, KRAS status and metastatic site. Although the results are potentially interesting and helpful, there are several issues to be addressed before considering its acceptance.

Major comments:

1.   Based on the authors’ description in the manuscript, median OS was 12.0 months (95% CI 10.2-13.9), while the figure 2 showed that the median OS of combination therapy seems to be much longer than 12 months. The authors should double check this data and address it.

2.    In the safety section, is there any new or worse safety issue of combination therapy observed compared to chemotherapy from historical data? If there is, the authors should add further explanations in the discussion section.

Minor comment:

1.     Please check the word spellings across the manuscript

For example, the spelling error of the word “eceived” should be corrected on Line 171, Page 4.

Comments on the Quality of English Language

Minor spelling errors should be corrected

Reviewer 2 Report

Comments and Suggestions for Authors

As we know, RCT also has its limitations, so not always the results of these studies coincide with the achievements of everyday practice. As a rule, highly qualified and specially trained researchers in specialized institutions with a long experience do RCT. This greatly purifies the results, but at the same time does not take into account the features of everyday routine clinical practice. That is exactly what the VELOUR's study is all about. In order to understand whether it is possible to interpolate its results into real life, it is very important additionally assess the applicability and reproducibility of the results in ordinary circumstances. From this point of view, the article of Polish authors is relevant and confirms the clinical benefit of the drug and the appropriateness of such treatment.

Author Response

Regarding Report 2, the investigators were provided the comments of the referee and agreed with them. They appreciate the idea of verifying the results of RCT treatment by everyday routine clinical practice, which concerns all patients enrolled according to registered indication of the medicinal product. They are looking forward to having an opportunity to present the results of their long term observation of the CRC patients published in the reputable medical journal. Especially, that the results suggest a positive aspect of PFS and OS.

Regarding the progress data (lines 190-191) there was mistakenly used  sentence which concerned the data on progression rate. This data was more adequately described in the previous sentence. The second  sentence was removed from the text. 

Round 2

Reviewer 1 Report

Comments and Suggestions for Authors The authors have fully addressed all the issues mentioned and the manuscript has been sufficiently improved to warrant publication in Cancers